# Phenotype of *Coxiella burnetii* Strains of Different Sources and Genotypes in Bovine Mammary Gland Epithelial Cells

**DOI:** 10.3390/pathogens11121422

**Published:** 2022-11-26

**Authors:** Katharina Sobotta, Katharina Bonkowski, Carsten Heydel, Klaus Henning, Christian Menge

**Affiliations:** 1Institute of Molecular Pathogenesis, Friedrich-Loeffler-Institut, Naumburger Strasse 96a, 07743 Jena, Germany; 2Institute for Hygiene and Infectious Diseases of Animals, Justus-Liebig-University (JLU), Frankfurter Strasse 85-89, 35392 Giessen, Germany; 3Institute of Bacterial Infections and Zoonoses, Friedrich-Loeffler-Institut, Naumburger Strasse 96a, 07743 Jena, Germany

**Keywords:** *Coxiella burnetii*, genotype, bovine udder, epithelial cells, host cell response

## Abstract

Despite the high prevalence of *C. burnetii* in dairy herds and continuous shedding via milk by chronically infected cows, bovine milk is not recognized as a relevant source of human Q fever. We hypothesized that the bovine mammary gland epithelial cell line PS represents a suitable in vitro model for the identification of *C. burnetii*-strain-specific virulence properties that may account for this discrepancy. Fifteen *C. burnetii* strains were selected to represent different host species and multiple loci variable number of tandem repeat analysis (MLVA) genotypes (I, II, III and IV). The replication efficiencies of all strains were similar, even though strains of the MLVA-genotype II replicated significantly better than genotype I strains, and bovine and ovine isolates replicated better than caprine ones. Bovine milk isolates replicated with similar efficiencies to isolates from other bovine organs. One sheep isolate (Cb30/14, MLVA type I, isolated from fetal membranes) induced a remarkable up-regulation of IL-1β and TNF-α, whereas prototypic strains and bovine milk isolates tended to suppress pro-inflammatory responses. While infection with strain Nine Mile I rendered the cells partially refractory to re-stimulation with *E. coli* lipopolysaccharide, Cb30/14 exerted a selective suppressive effect which was restricted to IL-6 and TNF-α and spared IL-1β. PS cells support the replication of different strains of *C. burnetii* and respond in a strain-specific manner, but isolates from bovine milk did not display a common pattern, which distinguishes them from strains identified as a public health concern.

## 1. Introduction

*C. burnetii*, a Gram-negative obligate intracellular bacterium, is the causative agent of Q fever, a widely distributed zoonosis caused by inhalation of as few as 1 to 10 *C. burnetii* particles [1]. The most common sources for transmission to humans are birth products of domestic small ruminants, as huge bacterial numbers (10^9^ bacteria/g tissue) are excreted during parturition [2]. Seroprevalence of *C. burnetii* in herds of goats and sheep is reportedly very high, i.e., 50 to 60% [3,4,5,6].

Even though cattle are not a recognized source of human Q fever in Europe, *C. burnetii* seroprevalence in cattle has been found to be up to 79% at herd level in Western Europe [5,7], with values increasing with the age of animals [8]. *C. burnetii* infections were reported to be associated with placentitis in cases of bovine abortion and detection of Coxiella-like organisms within trophoblasts [9]. In infertile dairy cattle with chronic endometritis, *C. burnetii* was detected intralesionally and intracytoplasmically in macrophages in the endometrium [10]. *C. burnetii* shedding scarcely and sporadically occurs in bovine feces, whereas 50% of cows were found to shed the agents by vaginal mucus intermittently or sporadically [11]. Asymptomatic, chronically infected cattle primarily shed *C. burnetii* in milk [12,13], and in higher numbers as compared to small ruminants [14]. Almost 40% of cows were detected as milk shedders with persistent and sporadic shedding patterns in one study [11]. Significantly higher estimated titers of *C. burnetii* were observed in cows with persistent shedding patterns, suggesting the existence of heavy-shedder cows [11]. *C. burnetii* shedding in milk was associated with chronic subclinical mastitis as measured by milk somatic cell counts [15]. These findings imply that *C. burnetii* exerts a tropism for the mammary glands in dairy cattle. The target cell in the mammary tissue has not been resolved in situ, but in vitro experiments have revealed that *C. burnetii* replicates in bovine mammary gland epithelial cells and even more efficiently than in epithelial cells from the placenta, lungs or intestine [16].

Studies from the US, Spain and Germany have shown that >94%, 72% and 63%, respectively, of tested bulk milk tank samples were PCR-positive for *C. burnetii* [17,18,19]. When testing shop and farm retail dairy products from Latvia, 26.67% of unpasteurized Latvian cow milk samples were PCR-positive, whereas 76.47% of pasteurized equivalents and 63.13% of fermented milk products were PCR-positive [20]. High *C. burnetii* numbers in bovine milk are of particular concern from a One Health aspect. Shedding in milk is considered to be associated with rapid dispersal of bacteria within a herd, e.g., by transmission from dams to calves via contaminated milk and colostrum [14,21,22]. Proposed by several authors as sources of human infection [23,24,25], the contribution of milk ingestion, mainly drinking unpasteurized milk, to Q fever infection in humans is difficult to establish [26]. Modern pasteurization protocols for high-temperature, short-time pasteurization of milk are effective in inactivating *C. burnetii* [27], reassuring the high safety level of off-the-shelf dairy products. However, in light of the marketing of dairy products produced from raw milk and increasing consumer preference for purchasing unpasteurized milk on the farm, the basis for the actual discrepancy between high *C. burnetii* prevalence in bovine bulk tank milk and the apparent low risk for humans must urgently be resolved.

Underlying reasons might be either genotypic or phenotypic peculiarities of the *C. burnetii* strains infecting cattle and colonizing the mammary gland, or host factors shaping the local environment for *C. burnetii* replication and release from host cells, or combinations thereof. Despite the high *C. burnetii* prevalence, genetic diversity of *C. burnetii* within a herd is rather low [28], and a single genotype of *C. burnetii* (ST20) is commonly found in US bovine milk samples [29,30], which suggests the existence of bovine udder-specific, low-virulence strains. However, ST20 is closely related to strain MST20, which was implicated in a recent outbreak in the Netherlands in 2011, where more than 4000 human cases were reported and approximately 52,000 ruminants were culled to control the outbreak [31]. Three different genotypes (MST20, MST33 and G) were isolated from humans and ruminants during that outbreak [32]. Strains isolated from humans can also be found in ruminants, even though the disease does not seem to be the same [32], whereas the US ST20 strain has a reduced ability to cause diseases in human or animals [29]. in vitro analyses with bovine and human macrophages indicated that *C. burnetii* replication is primarily determined by genotype; strains of MLVA genotype IV, such as Nine Mile Phase I (NMI) or the human strain Henzerling, replicate with a very high efficiency in cells of both species [33]. The public health risk in Belgium was linked to specific genomic groups, which were mostly found in small ruminant strains [34]. In France, MLVA clusters were found to be significantly associated with ruminant species, with all the cattle genotypes belonging to a “cattle-specific” cluster, whereas small ruminant genotypes were essentially grouped into two other clusters [35]. The results obtained by a functional assay deploying human peripheral blood mononuclear cells in turn suggested that cytokine responses are dependent on host origin rather than MLVA genotype [36].

Aiming at identifying markers for host adaptation and virulence of *C. burnetii* strains, we used the bovine mammary gland epithelial cell line PS as an in vitro infection model for the characterization of bacterial replication and host cell response after infection with 15 different *C. burnetii* isolates, which represented MLVA groups I to IV and were isolated from different hosts and organs, including three bovine milk isolates.

## 2. Materials and Methods

### 2.1. Cell Cultivation and Bacterial Strains

PS, a bovine mammary gland epithelial cell line from the secretory parenchyma [37], was kindly provided by P. Germon and P. Rainard (INRA, Université Tours, Nouzilly, France) and further characterized by our group [16]. Cells were cultured with advanced DMEM/Ham’s F12 nutrient mix [1:1] (Gibco, Life Technologies, Darmstadt, Germany) supplemented with Insulin-like growth factor (10 ng/mL; Peprotech, Hamburg, Germany), Fibroblast growth factor (5 ng/mL; Peprotech), Epidermal growth factor (5 ng/mL; Sigma-Aldrich, Taufkirchen, Germany), Hydrocortisone (1 µg/mL, Sigma-Aldrich), HEPES buffer 20 mM (Fisher Scientific, Schwerte, Germany) and L-Glutamine (2 mM, Life Technologies; [37]. Maintenance medium was supplemented with 10% fetal bovine serum (FBS; Fisher Scientific; heat-inactivated (56 °C for 30 min)). The test medium used in the inoculation experiments and cytokine analysis contained 1% FBS. Cells were seeded into 24-well plates in numbers of 4–10 × 10^4^ and cultured for 2 to 3 days to reach an 80% confluent cell monolayer.

The *C. burnetii* strains employed in the study are listed in Table 1. Unless otherwise indicated, strains were from the collection of the Institute for Hygiene and Infectious Diseases of Animals (Justus-Liebig-University, Giessen, Germany). Strains Cb23/2, Cb71/3, Cb98/2, Cb19/34, Cb30/14, Cb28/6 and Cb33/4 were isolated by the German National Reference Laboratory for Q fever (Jena, Germany). All strains were isolated from different animals. MLVA genotyping was conducted using flanked primers for seven previously described microsatellites [38]. The plasmid status of *C. burnetii* strains was determined by multiplex PCR [33,38,39]. The inoculum of strain Z69/06, originally genotyped as type III [39], was quality controlled in retrospect and then presented as a mixed culture of 2 *C. burnetii* strains with different genotypes (III and IV). The preparation of the inocula was performed as already described [40]. In brief, *C. burnetii* aliquots were prepared with 1 × 10^9^ bacterial cells per ml NaCl solution, stored at −80 °C, thawed at ambient temperature and resuspended thoroughly for at least 1 min to separate bacterial agglutination immediately prior to inoculation [40].

### 2.2. C. burnetii Replication Kinetics

PS cells were inoculated at a multiplicity of infection (MOI) of 100 by the addition of *C. burnetii* suspensions in NaCl solution to test medium. After inoculation at 37 °C and 5% CO_2_ for 24 h (1 day post-inoculation, d p.i.) and at 3, 7 and 14 d p.i., cell monolayers were washed 3 times with pre-warmed PBS (37 °C) and replenished with test medium. Triplicate cell cultures were harvested at different time points to monitor replication efficiency. To this end, cells were washed three times with warm PBS and detached with Trypsin/EDTA solution. To release bacteria from inside the cells, the cells were frozen (−80 °C) and thawed three times and subsequently incubated for 30 min at 95 °C to inactivate the bacteria.

DNA of the infected cell cultures was purified with the Invisorb^®^ DNA Cleanup-Kit (Stratec, Birkenfeld, Germany), according to the manufacturer’s instructions. The number of genome equivalents (GEs) was monitored by quantitation of the isocitrate dehydrogenase (*icd*) gene by quantitative real-time PCR (qPCR) [40,41].

A replication factor was calculated by dividing the log10 values for GEs detected at 14 d p.i. by the respective values detected at 1 d p.i.

### 2.3. Determination of C. burnetii-Induced Host Cell Response

PS cells were inoculated as described above and harvested either at 1 or 7 d p.i. Control cells were mock inoculated with NaCl solution (negative controls) or stimulated with *E. coli* O111:B4 LPS (5 µg/mL; Sigma-Aldrich) as positive controls. For LPS re-stimulation, cell culture medium of infected PS was supplemented with *E. coli* LPS (5 µg/mL) at 7 d p.i., and cultures were incubated for a further 4 h at 37 °C and 5% CO_2_. In some experiments, cells were challenged with a heat-killed suspension of NMI (95 °C, 10 min) with a theoretical MOI of 100. Thereafter, the cells were washed with PBS and subsequently lysed with RLT buffer (RNeasy Mini Kit; Qiagen, Hilden, Germany). Total RNA was isolated with an RNeasy Mini Kit (Qiagen), according to the instructions of the manufacturer. To avoid DNA contamination, RNA was purified with the RNase-free DNase set (Qiagen).

### 2.4. Reverse Transcription and Cytokine-Specific Real-Time PCR

Equal RNA amounts from each sample were reverse transcribed into cDNA, as described previously [40]. Relative gene expression levels of different host-specific cytokines in comparison to GAPDH as a housekeeping gene were determined by quantitative real-time SYBR Green-based (Applied Biosystem, Waltham, MA, USA) PCR, using ABI Prism^®^7500 (Applied Biosystem). Reactions with all primers (Table 2) were run at an annealing temperature of 60 °C. The PCR reaction profile was: denaturation (15 s, 95 °C) followed by annealing (1 min, 60 °C; 39 cycles) and a melting step (15 s, 60 °C). Relative gene expression levels were calculated using the relative expression software REST [42].

### 2.5. Statistical Analysis

Statistical comparisons were conducted using *t*-tests or Mann–Whitney U tests with the statistical software XLSTAT. A *p*-value of ≤0.05 (“a” or “*”) indicated a statistically significant difference at the 95% confidence level; a *p*-value of ≤0.01 (“b” or “**”) indicated a statistically significant difference at the 99% confidence level. Real-time PCR data were analyzed by a randomization test with pairwise reallocation (software REST [42]).

## 3. Results

### 3.1. Different C. burnetii Strains Replicate with Similar Efficiencies in Bovine Mammary Gland Epithelial Cells

After the addition of the *C. burnetii* suspension to monolayers of PS cells, bacterial numbers in the cultures were monitored for each strain over a culturing period of 14 d (Figure 1). The growth curves for all *C. burnetii* strains, except strains Nine Mile I, Henzerling and Scurry, in these bovine mammary gland epithelial cells were characterized by an initial lag-phase without replication or even with a decrease in genome equivalent (GE) numbers within the first 3 days post-inoculation (d p.i.). Thereafter, the GE numbers of all the assessed strains significantly increased by two to four orders of magnitude until 14 d p.i., with some variation between strains.

To compare the replication efficiencies between the strains, a replication factor was calculated by the relative increase in GEs between days 1 and 14 p.i. Since replication of *C. burnetii* in bovine macrophages was found to vary with MLVA genogroup [33], replication factors of the strains in PS cells were grouped according to the strains’ genotypes (Figure 2a). Differences between genotypes were only marginal, although strains of MLVA genotype II replicated significantly better than genotype I strains.

Bovine isolates of *C. burnetii* are more difficult to obtain by culturing approaches than isolates from small ruminants (Henning, personal observation). Isolates from cattle and sheep included in this study replicated significantly better in bovine mammary gland epithelial cells than caprine isolates, but their replication rates were not significantly different from each other (Figure 2b).

As compared with epithelial cells from other outer surface linings of cattle (lung, gut, placental), *C. burnetii* strains Nine Mile I and II exhibit a prime tropism for bovine mammary gland epithelial cells in vitro [16]. To test whether bovine strains isolated from milk (Z69/06, Cb33/4 and Cb28/6) replicate significantly better in PS cells than bovine strains isolated from tissues (Cb98/2 and Z488/94), replication efficiencies were compared but found to not significantly differ (Figure 2c).

### 3.2. C. burnetii Infection Barely Affects Host Response in Bovine Mammary Gland Epithelial Cells

Different from observations with bovine macrophages [40], *C. burnetii* induced no early activation of a pro-inflammatory immune response in bovine mammary gland epithelial cells. Neither of the strains induced detectable IL-1β, IL-6 and TNF-α host cell responses at 1 d p.i. (data not shown). Even 7 days after inoculation with *C. burnetii*, differences in the expression of pro-inflammatory cytokines compared with uninoculated PS cells were only barely detectable (Figure 3). Only strain Cb30/14 stood out due to its particular potency in inducing an up-regulation of the transcription of genes encoding IL-1β and TNF-α. Transcription of these genes was significantly reduced by the strain Scurry. While the latter strain in turn significantly induced IL-6 transcription, milk isolates Z69/06, Cb28/6 and Cb33/4 showed a tendency to depress IL-6-transcription to a similar degree as the prototype strain NMI. Bovine strains from other organs (Cb98/2 and Z488/94) resulted in no changes in cytokine expression compared to cell controls. Unlike *C. burnetii*-infected bovine macrophages [40], pro-inflammatory host responses in bovine mammary gland epithelial cells did not correlate with the *C. burnetii* MLVA genotype.

### 3.3. C. burnetii Blocks Pro-Inflammatory Responses of Bovine Mammary Gland Epithelial Cells in a Strain-Specific Manner

The scarce response of cells to infection can be taken as evidence that *C. burnetii* sustainably inhibits cell response to promote its replication in the host cell. To prove this hypothesis, we exposed bovine mammary gland epithelial cells to the reference strain NMI or a heat-inactivated equivalent for 24 h, cultured the cells for a further 7 days and challenged the infected cells with *E. coli* LPS. NMI significantly inhibited *E. coli* LPS-induced expression of IL-1β, IL-6 and TNF-α (Figure 4a). Different from viable NMI, the heat-inactivated NMI suspension could not inhibit the cytokine expression following *E. coli* LPS exposure. Different from NMI, strain Cb30/14, which was found to significantly up-regulate expression of IL-1β and TNF-α at 7 d p.i. (Figure 3), inhibited LPS-triggered expression of TNF-α but not that of IL-1β induced by *E. coli* LPS (Figure 4b).

## 4. Discussion

A recent bibliometric analysis [43] revealed that an article on a comparison of *C. burnetii* shedding in the milk of dairy bovine, caprine and ovine herds [14] was the most highly cited in the current literature on *C. burnetii*. The study demonstrated that the bacterium is mainly excreted through the milk of infected cattle and goats, while in sheep it is mainly excreted through feces and vaginal excretions [14]. Indeed, the findings may explain the higher association of human outbreaks with sheep as compared to cattle and goats. However, the events leading to or preventing transmission of *C. burnetii* from cattle to humans have not been adequately investigated. Preceding in vitro studies with a bovine mammary gland epithelial cell line had demonstrated that prototypic *C. burnetii* strains NMI and II replicated in these cells [16]. We therefore studied replication and host cell response to a selection of *C. burnetii* strains of different genotypes and from different sources, including three bovine milk isolates, in the PS cell model.

The different *C. burnetii* strains all replicated in bovine mammary gland epithelial cells with high efficiency. The possibility of genuine replication of *C. burnetii* in bovine mammary gland tissue is consistent with reports that the bulk tank milk of dairy herds in the US and Europe contains substantial numbers of *C. burnetii*, e.g., more than 10^2^ bacteria/mL [18,19,44,45]. The differences in replication efficiencies that we observed in the present study were minor, even though strains of the MLVA genotype II, on average, replicated significantly better than genotype I strains. This deviates from findings with bovine monocyte-derived macrophages (MDMs), in which replication rates varied mainly between MLVA genotypes III and IV [33]. While human MDMs supported the replication of different *C. burnetii* strains more equally, bovine MDMs particularly supported the propagation of strains NMI, Henzerling and Scurry. The strain Dugway, as well as isolates from ruminants, rather poorly replicated in the cells. This pattern is reminiscent of the pattern observed in this study for bovine mammary gland epithelial cells, as the growth curves of the latter strains were characterized by a prominent delay in replication, which started only after 3 d p.i. However, this pattern was shown by all more recent *C. burnetii* isolates, and the apparent genotype restriction of *C. burnetii* bovine milk isolates does not appear, therefore, to result from the selective propagation of certain strains by tissue-specific cells. This study also revealed that, on average, bovine and ovine isolates replicated better in PS cells than caprine isolates, which correlates with the primary *C. burnetii* shedding route of the animals [14]. We had hypothesized earlier that bovine macrophages serve in bacterial transport from the entry site in the body, which might be the airways or the intestinal mucosa, to target cells for bacterial replication at exit sites, such as epithelial cells, in different organs [16]. Bovine MDMs are less supportive of *C. burnetii* replication than PS cells by approximately one order of magnitude [16,33]. Nevertheless, it is tempting to speculate that selection for bacterial host adaptation and virulence starts early in the infection process, i.e., inside the initial *C. burnetii* target cells, such as lung macrophages and dendritic cells, which was not mimicked by the model applied herein. Co-culturing models or inoculation of mammary gland epithelial cell cultures with infected macrophages may be needed to properly represent all aspects of this process.

Another set of determinants to be considered that are instrumental for limiting *C. burnetii* replication and subsequent shedding in milk are the local inflammatory responses after some rounds of replication in tissue-specific cells. Indeed, *C. burnetii* shedding was linked to subclinical mastitis, with elevated levels of somatic cell counts indicative of some inflammation upon infection of the organ [15]. Epithelial cells constitute the first line of defense against microbial pathogens, act as a barrier for bacterial recognition and use their immunoregulatory function to alert the immune system to reduce immersive pathogens [46]. Interactions with pathogens usually induce host responses with an up-regulation of different inflammatory factors, e.g., cytokines, chemokines or cell-associated surface markers. Bovine MDMs respond to *C. burnetii* infection with an early (3 h p.i.) increase in the transcription of pro-inflammatory cytokine genes, such as those encoding for IL-1β, IL-12 and TNF-α [40]. In contrast, bovine mammary gland epithelial cells showed no immune response after infection with the reference strains NMI and NMII during the first 24 h p.i. [16]. Similar results were obtained herein after infection with different *C. burnetii* strains. Attachment of *C. burnetii* to surface PAMP receptors, such as TLR2 [47], seemingly fails to induce pro-inflammatory factors in udder cells. Moreover, some strains suppressed inflammatory gene expression at 7 d p.i., i.e., when cells were analyzed after the onset of *C. burnetii* replication. Re-stimulation studies with *E. coli* LPS and *C. burnetii*-infected PS cells revealed that the lack of inflammatory cellular response results from a refractory state rather than from lack of recognition of pathogen-associated molecule patterns (PAMPs). Although the early induction of IL-1β in bovine MDMs by the strain NMI is independent of a functional bacterial metabolism [40], the refractory state of infected PS cells was found to be an actively controlled process of *C. burnetii*, since cells exposed to heat-inactivated strain NMI suspensions reacted like uninfected controls. Previous reports show that Coxiella inhibit the activation of apoptosis in host cells to promote cell viability [48]. Three anti-apoptotic effector proteins (AnkG, CaeA and CaeB) have been identified [49,50,51] and are transported to the cytosol via a type IV secretion system [52]. Specific virulence factors may also be deployed by *C. burnetii* to manipulate the host response to create a safe replicative niche in bovine udder cells. Cross-signaling to inflammatory response genes as an explanation of the anti-inflammatory effects of *C. burnetii* in bovine cells deserves further investigation.

Of note, a single isolate, Cb30/14, obtained from sheep, induced a significant increase in cytokine gene transcription, e.g., IL-1β and TNF-α. Increased concentrations of pro-inflammatory cytokines were found in milk from cows with coliform mastitis [53]. Mastitis in dairy herds of cattle [15] is associated with shedding of *C. burnetii* in milk. Secretion of pro-inflammatory cytokines stimulates the migration of somatic cells and neutrophils to the udders [53]. In bovine MDMs, infection with isolate Cb30/14 resulted in a mitigated general cellular response but a selective increase in IL-1β and IFN-γ expression [33]. IFN-γ promotes an up-regulation of major histocompatibility complex (MHC) expression in epithelial cells [54]. In Chlamydia-infected epithelial cell lines, increased MHCI expression promotes the degradation of the pathogen by activation of CD8^+^ cytotoxic T-cells [55]. Earlier findings by our group, however, called into question whether the increased amounts of mRNA after *C. burnetii* infection of bovine cells are translated into proteins. During the early phase of infection of bovine MDMs, *C. burnetii* induces a marked increase in IL-1β mRNA but fails to increase mature bioactive IL-1β [40]. Different from the situation in bovine MDMs, infected epithelial cells respond to Cb30/14 infection rather late, i.e., days after the invasion process, and therefore presumably do not engage cell surface receptors. Activation in *C. burnetii*-infected mammary gland epithelial cells probably occurs instead via receptors located inside parasitophorous vacuoles, such as Toll-like receptors (TLRs) 3, 7, 8 and 9. Endosomal TLR3 plays a critical role in host immune response in Chlamydia-infected epithelial cells [56]. Although all other *C. burnetii* strains acted differently but were similar to each other, the exceptional findings for strain Cb30/14 imply that some differences in host cell–pathogen interactions do occur at the main exit site for *C. burnetii* in cattle.

The genotypes of the selected *C. burnetii* isolates have been defined only at the level of MLVA genotype using seven previously described microsatellites [38,39] for the purpose of this study. A harmonized reference method for the molecular characterization of *C. burnetii* has yet to be designed. The most commonly adopted methods to define phylogeny are multi-loci variable-number tandem repeat analysis (MLVA) and multispacer sequence typing (MST), with MLVA being more discriminatory than MST [34]. When attempting to dissolve the phylogeography of human and animal strains by genetic fingerprinting in Belgium, the high discriminative power conferred by the MLVA method performed with 13 markers allowed the definition of three MLVA clusters divided into 23 subclusters, which was crucial for in-depth genetic analysis [34]. Deploying more markers than were used in our study may also help to better link genotypes to phenotypes, even though more isolates would have to be included to represent each subcluster in sufficient numbers to confirm correlations.

The virulence of *C. burnetii* differs between strains, apparently because of isolate-specific genes, pseudogenes [57] or plasmid types [58]. In a rodent model of acute Q fever infection, the virulence of *C. burnetii* was associated with bacterial genotype. Strains with the same genotype cause the same pathology in guinea pigs and the same cytokine secretion in response to *C. burnetii* infection in a mouse model [59]. To date, more than 130 putative effector proteins have been identified [60], some of which have anti-apoptotic activity [50,51,52], whereas the majority still await functional characterization [49]. In a selection of different *C. burnetii* strains, pseudogenes were found that could be attributed to pathotype-specific virulence of Coxiella [57]. Moreover, *C. burnetii* isolates from raw bovine milk, uterus swab samples from dairy cattle with reproductive disorders, aborted bovine fetus samples and mammary gland samples from healthy dairy cattle had various degrees of pathogenicity for guinea pigs, even though the protein and lipopolysaccharide (LPS) profiles of these strains were similar to those of the reference strain of phase I [61]. A bovine *C. burnetii* isolate was found to multiply faster than goat isolates in a bovine macrophage cell line, pointing to a preferential specificity of this strain for homologous host cells [62]. Identification of particular genetic determinants encoding for host adaptation and virulence was beyond the scope of this study. The findings obtained herein add important aspects to our understanding of the interactions of *C. burnetii* with bovine hosts, in that it is evident that beyond host of isolation and (MLVA) genotype, the type of the incremented host cell is another variable to be considered in assessing *C. burnetii* isolates as to their level of host adaptation and virulence. Moreover, preliminary data from our laboratory indicate that cells from cattle and small ruminants interact differentially with *C. burnetii* organisms (data not shown).

There are indications that *C. burnetii* isolates possess some degree of adaptation to the host species they mostly circulate in and only occasionally become transferred to an aberrant host, e.g., humans [34,36]. We therefore hypothesized that bovine *C. burnetii* isolates from milk are more adapted to their host organs than other bovine isolates. Our results show that milk isolates replicated with nearly the same efficiency in bovine udder epithelial cells than bovine isolates with other origins. Consequently, the distribution of *C. burnetii* inside the host is isolate-independent, and organotropism cannot be confirmed. Some genetic diversity of *C. burnetii* isolates was also described in different Spanish bulk tank samples [28]. Separated isolates are closely related to isolates from human outbreaks and persist inside a herd over a long time [28]. Thus, the milk of infected animals represents a risk factor for the spread of *C. burnetii* infection within a flock, apparently independently of *C. burnetii* isolate genotype.

## 5. Conclusions

In conclusion, we have illustrated for the first time the infection course of different *C. burnetii* isolates in bovine mammary gland epithelial cells. Different *C. burnetii* strains replicated with similar efficiencies in bovine udder cells without stimulating host cells. These results suggest that mammary gland epithelial cells could be a replicative niche for *C. burnetii* infection in cattle. The reported ubiquitous distribution of high *C. burnetii* seroprevalence in cattle herds and the high *C. burnetii* concentration in bulk tank milk, together with our findings that bovine milk isolates did not display a common pattern that distinguished them from strains identified as of public health concern, point to unpasteurized bovine milk as another potential source of zoonotic human Q fever.

## Figures and Tables

**Figure 1 pathogens-11-01422-f001:**
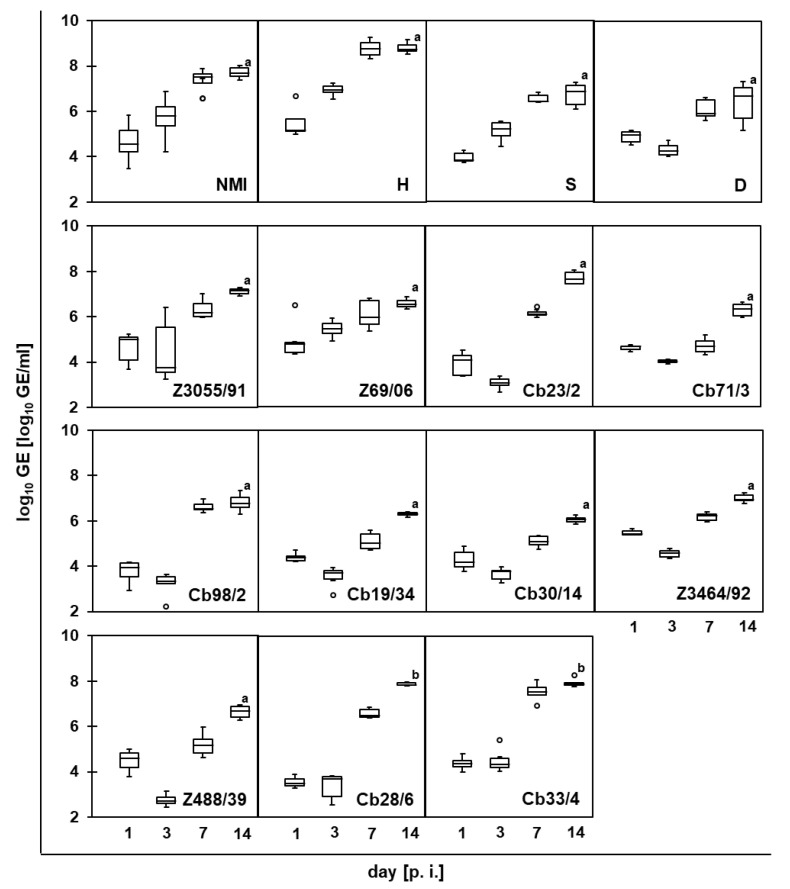
Replication of *C. burnetii* strains in bovine mammary gland epithelial cells. PS cells were inoculated with *C. burnetii* strains (strain designations as listed in Table 1 given in bold in all panels; 100 MOI, 24 h), and genome equivalents (GEs) were quantified by *icd* real-time PCR at 1 d (i.e., 24 h after addition of the *C. burnetii* suspension) and at 3, 7 and 14 days post-inoculation (d p.i.). GE values determined in technical duplicates in four independent experiments per strain are depicted as box plots (significantly different to 1 d p.i.: a = *p* ≤ 0.01; b = *p* ≤ 0.05).

**Figure 2 pathogens-11-01422-f002:**
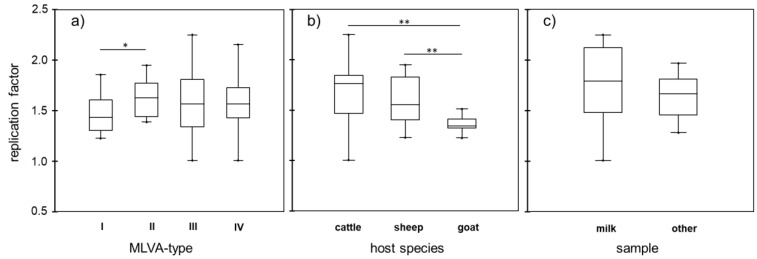
Replication efficiencies of *C. burnetii* strains in bovine mammary gland epithelial cells grouped by MLVA group (**a**), by host species (**b**) and by source sample (**c**) (only bovine isolates included in the latter panel). Increases in genome equivalents (GEs) between 1 and 14 days post-inoculation are calculated as replication factors and presented as mean values of technical duplicates from four independent experiments per strain. Mean values per group are depicted as black lines (* *p* ≤ 0.05; ** *p* ≤ 0.01).

**Figure 3 pathogens-11-01422-f003:**
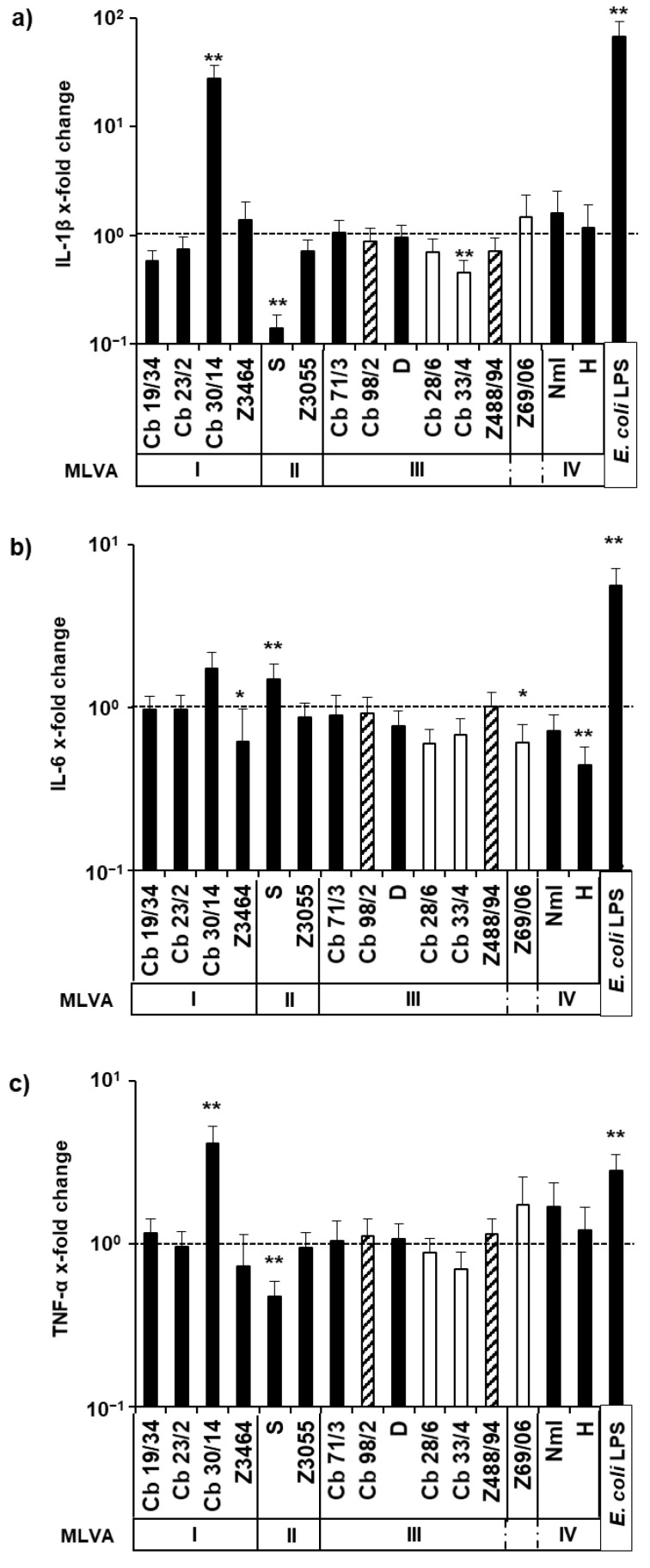
Relative amounts of cytokine-specific mRNA molecules in *C. burnetii*-infected mammary gland epithelial cells. PS cells were inoculated with *C. burnetii* strains (strain designations as listed in Table 1 given in bold in all panels; 100 MOI, 24 h). Amounts of mRNA encoding for IL-1β (**a**), IL-6 (**b**) and TNF-α (**c**) were quantified at 7 days post-inoculation, normalized to GAPDH and calculated relative to cell controls (fold increase). The results of four independent experiments per strain, each with technical duplicates, are depicted. The results for different strains are arranged according to MLVA group (open bars indicate bovine milk isolates, hatched bars indicate other bovine isolates; *: *p* ≤ 0.05; **: *p* ≤ 0.01).

**Figure 4 pathogens-11-01422-f004:**
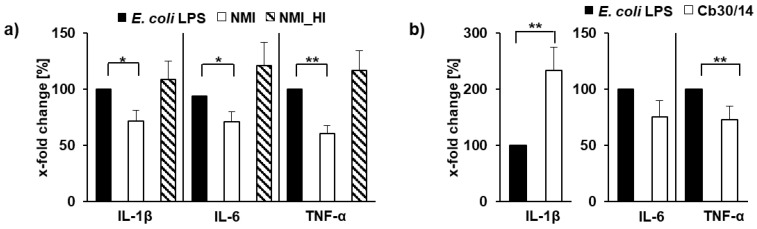
Host response of *C. burnetii*-inoculated bovine mammary gland epithelial cells to re-stimulation with *E. coli* LPS. PS cells were left uninfected, were inoculated with *C. burnetii* strain Nine Mile I (NMI) or exposed to a heat-inactivated NMI suspension (**a**) or inoculated with strain Cb30/14 (**b**). At 7 days post-inoculation, cultures were re-stimulated with *E. coli* LPS for a further 4 h. Amounts of cytokine-specific mRNA (IL-1, IL-6 and TNF-α) quantified in four independent experiments per strain, each with technical duplicates, are depicted relative to uninfected, *E. coli* LPS-re-stimulated control cells (values set to 100%; *: *p* ≤ 0.05; **: *p* ≤ 0.01).

**Table 1 pathogens-11-01422-t001:** *C. burnetii* strains used in this study.

Strain	Abbr.	Genetic Properties				Q Fever/Coxiellosis			Origin	
		MLVA Genotype	Plasmid	*adaA*		Disease	Course		Species	Sample
Nine Mile I (493) ^1^	NMI	IV	QpH1	+		U.	U.		Tick	U.
Henzerling	H	IV	QpH1	+		Pneumonia	Acute		Human	U.
Scurry	S	II	none	−		Hepatitis	Chronic		Human	U.
Dugway ^2^	D	III	QpH1	+		U.	U.		Rodent	U.
Z3055/91		II	QpH1	+		U.	U.		Sheep	Vaginal swab
Z69/06		III	QpH1	+		U.	U.		Cattle	Milk
Cb23/2		I	QpH1	+		U.	U.		Sheep	Fetus
Cb71/3		III	QpH1	+		U.	U.		Goat	Fetal membranes
Cb98/2		III	QpH1	+		U.	U.		Cattle	Unspecified tissue
Cb19/34		I	QpH1	+		U.	U.		Goat	Fetus/fetal membranes
Cb30/14		I	QpH1	+		U.	U.		Sheep	Fetal membranes
Z3464/92		I	QpH1	+		Abortion	U.		Goat	Fetal membranes
Z488/94		III	QpH1	+		Abortion	U.		Cattle	Fetal membranes
Cb28/6		III	QpH1	+		U.	U.		Cattle	Milk
Cb33/4		III	QpH1	+		U.	U.		Cattle	Milk

U. = Unknown. ^1^ Nine Mile I was included as a reference strain only. Data obtained for Nine Mile I in this manuscript have been published previously [16]. ^2^ Dugway strain 5J108-11 reportedly contains a QpDG plasmid. However, of the 3 strains of Dugway available, we used a strain which did not carry the QpDG plasmid.

**Table 2 pathogens-11-01422-t002:** Sequences of primers used for cytokine-specific real-time PCR in this study.

Primer	Sequence 5′-3′
GAPDH	F: GCG ATA CTC ACT CTT CTA CCT TCG A
	R: TCG TAC CAG GAA ATG AGC TTG AC
IL-1β	F: ACC TGA ACC CAT CAA CGA AAT G
	R: TAG GGT CAT CAG CCT CAA ATA ACA
IL-6	F: CTG AAG CAA AAG ATC GCA GAT CTA
	R: CTC GTT TGA AGA CTG CAT CTT CTC
TNF-α	F: TCT TCT CAA GCC TCA AGT AAC AAG T
	R: CCA TGA GGG CAT TGG CAT AC

## Data Availability

The data presented in this study are available from the corresponding authors on request.

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
