# Peer review of "Phenotype of Coxiella burnetii Strains of Different Sources and Genotypes in Bovine Mammary Gland Epithelial Cells"

_pathogens, 2022, doi:10.3390/pathogens11121422_

Round 1

Reviewer 1 Report

Brief Summary:

This paper describes a study of the infection course and cellular immune responses induced by different (MLVA genotype, host species and sample origin) strains of Coxiella burnetii in bovine mammary epithelial cells with the aim of furthering the understanding the specific role bovine mammary gland infection might play in zoonotic infection of humans.

General Comments

The study is novel and builds on previous knowledge in this area by providing insight into the pathogenesis of C. burnetii infection in mammary epithelial cells; an area that remains enigmatic and poorly understood. Understanding this is very important from a public health perspective due to the ongoing ingestion of unpasteurized milk and the potential for aerosol transmission to humans in milk in dairy parlours. It is also important at the animal level because it represents a pathway of shedding and transmission among cattle within herds.

While the manuscript is generally well written, I think it could benefit from some assistance with the English to improve the clarity in some areas and make the paper easier to read and understand. 

Specific Comments:

Abstract

Lines: 12-13 – rather than the word ‘permanent’ perhaps the word ‘ongoing’ could be considered

Lines 13-14: While it is apparently true in European countries that cattle are rarely associated with human Q fever, this is not the case in all countries world-wide. As an example, see the following reference: Graves SR, Islam A. Endemic Q Fever in New South Wales, Australia: A Case Series (2005-2013). Am J Trop Med Hyg. 2016 Jul 6;95(1):55-9. doi: 10.4269/ajtmh.15-0828. Epub 2016 May 2. PMID: 27139451; PMCID: PMC4944709.

Introduction

Throughout the introduction and particularly this paragraph, general statements are made relating to a particular study however the statements should clarify that these results are to a specific study or studies and not a general fact across all herds or animals. An example of this is in lines 47-48 which states: “Almost 40% of cows were detected as milk shedders with persistent and sporadic shedding patterns.”

Line 33: include the words “as little as” between “inhalation of” and “1 to 10 C. burnetii particles”.

Line 38: change “no” to “not a” and also see comment above re lines 13-14 about this being a European phenomenon and not necessarily a global finding. 

Line 38: Also change the word ‘no’ to ‘not’.

Line 40: suggest changing the word ‘were’ to ‘may be’.

Line 57-59: where these Latvian samples positive via PCR. Should acknowledge that although DNA may be present in pasteurized samples the organisms may not be viable and therefore may not represent a public health risk.

Lines 63-65: the grammar of this sentence could be improved for clarity.

Line 71: do you mean “resolved” rather than “dissolved” here?

Line 73: what is meant by “settling in the mammary gland”?

Line 78: Would the words “was implicated” be better than “participated” here?

Line 79-80: 4,000 rather than 4.000 and 52,000 rather than 52.000

Line 80: change the words “had to be” to “were” as some commentators say they could have used vaccination of both humans and animals more to prevent the large numbers of animals being culled so maybe they didn’t have to be culled.

Lines 82-83: Given this is a zoonotic disease and transmission rarely occurs from humans to other humans I think talking about “human strains” is strange. It is likely that isolates obtained from humans are also present in animals.

Lines 95-99: If this is the aim of the present study it should be indicated as the aim of the present study somewhere in this paragraph.

 Results

Lines 113-117 Figure legend (check this for all other figures too) – include explanation of p. i., NMI, S, D and H as the figures should stand alone from the text.

Lines 151-153 – here and at other places in the manuscript, isolates are described as coming from milk or other organs however it is unclear whether this is different isolates from the same animal or isolates obtained from different animals. In my experience, I have obtained the same genotype from multiple site within the same animal (e.g. milk, placenta, vaginal swab etc).

Discussion

Line 208 to 209: Is the reference for this statement still 33 as in lines 205-206? This is unclear here.

Lines 215-217: why is bovine isolates replicating better in the PS cells than caprine isolates in contrast with the primary shedding route in animals given in lines 46 and 47 you state that cattle shed primarily in milk and in higher numbers than small ruminants?

Lines 229-231: reference needed.

Lines 313-315: is this “data not shown”?

Table 1

Correct spelling of swab in sample column

 Conclusion

Line 409: I believe this may be the first use of acronym: MEC so write in full

Line 411: Suggest use "ubiquitous" instead of "ubiquitary"

Line 413: this is very awkward grammar.

Author Response

Please see attached rebuttal letter for the response to all three reviews.

Reviewer 2 Report

The paper presents interesting results but it raises some questions about why the authors used the genogroup MLVA instead Hendrix genogroup normally used by other authors. It allows better analysis and comparison with other works.

Author Response

(The authors gave the same response as above.)

Reviewer 3 Report

The manuscript by Sobotta et al., entitled “Phenotype of Coxiella burnetii strains of different sources and genotypes in bovine mammary gland epithelial cells” describes growth attributes and cytokine induction by 15 strains of C. burnetii, representing four genotypes, in a bovine mammary gland epithelial cell line (PS). The results are based on thorough analyses and the conclusions sound. In addition, they are timely given the prevalence of this pathogen in milk. However, there are a few items that need attention.

Major points-

1)      There are numerous English grammatical / syntax errors in the text (see minor points below).

2)      Figure 4b- white bars are labeled “11”, but the experiment used Cb30/14 C. burnetii; I’m not sure where this number comes from (perhaps it should read 14?). I am also curious why a heat-inactivated control was not included here, as in the experimental data shown in Fig. 4a.

3)      Lines 270-271- “Bovine MDM are…replication than PS cells”. Please cite the reference for the MDM results (perhaps reference 33?).

4)      Lines 257-258- Authors state that IL1-beta, IL-6 and TNF-alpha genes were significantly increased in transcription as a result of Cb30/14 infection, but IL-6 is not marked as significant in Fig. 3b.

5)      Figure 2- consider placing label “a)” into the corresponding panel to be consistent with panels b) and c).

Minor points-

1)      Lines 12 and 32- delete “C. burnetii” (this is a conventional abbreviation after the first use of an organism’s full name in the text).

2)      Line 20- replace “same” with “similar”.

3)      Lines 26 and 126- should read “strain-specific”

4)      Line 28- replace “of” with “a”

5)      Line 38- should read “…are not a recognized…”

6)      Line 41- should read “…abortion with visible Coxiella-like….”

7)      Line 46- should read “…chronically-infected…”

8)      Line 69- should read “…on the farm…”

9)      Line 71- replace “dissolved” with “resolved”

10)  Line 76- should read “…a herd is…”

11)  Line 79- should read “Netherlands in 2011...”

12)  Line 80- replace “notified” with “reported”

13)  Line 85- replace “humane” with “human”

14)  Lines 93-94- should read “…are dependent on host origin rather than…”

15)  Line 98- delete “either of”

16)  Line 108- replace “post inoculationem” with “post-inoculation”

17)  Line 147- should read “cytokines compared to uninoculated…”

18)  Line 152- should read “IL-6” (no italics) as in previous line

19)  Lines 158, 273- should read “C. burnetii-infected”

20)  Line 170- should read “…for a further…”

21)  Line 174- replace “exceptionally” with “significantly”

22)  Line 218- replace “for” with “in”

23)  Line 224- should read “…which were not…”

24)  Line 232- should read “…act as a barrier…”

25)  Line 235- should read “cell-associated”

26)  Line 236- replace “of” with “in”

27)  Line 244-245- replace “unveiled” with “revealed”

28)  Line 245- should read “…response results from a refractory state rather than…”

29)  Lines 252-253- should read “…and are transported to the host cytosol…”

30)  Line 256- should read “deserves further infestigtations.”

31)  Line 261- replace “stimulate” with “stimulates” (refers to secretion)

32)  Lines 265, 275- should read “Chlamydia-infected”

33)  Line 270- should read “Different from the situation…”

34)  Line 272- should read “and are therefore…”

35)  Line 277- no comma after “imply”

36)  Line 280- should read “The genotypes of the selected C. burnetii isolates have…”

37)  Line 317- should read “…transferred to an aberrant…”

38)  Line 321- replace “origin” with “origins”

39)  Line 322- should read “isolate-independent”

40)  Line 324-325- replace “persistent” with “persist”

41)  Line 326- should read “…increased risk factor…”

42)  Line 343- replace “deployed” with “employed”

43)  Table 1, column 9- “vaginal swab” should be “vaginal swab”;  delete period after “tissue”

44)  Lines 366, 368- add degree symbol for temps

45)  Line 381- replace “descripted” with “described”

46)  Line 392- replace “reversely transcribed” with “reverse-transcribed”

47)  Line 409- define “MEC” or spell out

48)  Line 411- replace “ubiquitary” with “ubiquitous”

49)  Line 413- should read “the not-to-be-underestimated” (compound adjective)

Author Response

(The authors gave the same response as above.)
